# SumVg: Total Heritability Explained by All Variants in Genome-Wide Association Studies Based on Summary Statistics with Standard Error Estimates

**DOI:** 10.3390/ijms25021347

**Published:** 2024-01-22

**Authors:** Hon-Cheong So, Xiao Xue, Zhijie Ma, Pak-Chung Sham

**Affiliations:** 1School of Biomedical Sciences, The Chinese University of Hong Kong, Shatin, Hong Kong, China; 1155151623@link.cuhk.edu.hk (X.X.); zhijiema@link.cuhk.edu.hk (Z.M.); 2KIZ-CUHK Joint Laboratory of Bioresources and Molecular Research of Common Diseases, Kunming Institute of Zoology and The Chinese University of Hong Kong, Shatin, Hong Kong, China; 3Department of Psychiatry, The Chinese University of Hong Kong, Shatin, Hong Kong, China; 4CUHK Shenzhen Research Institute, Shenzhen 518057, China; 5Margaret K. L. Cheung Research Centre for Management of Parkinsonism, The Chinese University of Hong Kong, Shatin, Hong Kong, China; 6Hong Kong Branch of the Chinese Academy of Sciences Center for Excellence in Animal Evolution and Genetics, The Chinese University of Hong Kong, Shatin, Hong Kong, China; 7Brain and Mind Institute, The Chinese University of Hong Kong, Shatin, Hong Kong, China; 8Department of Psychiatry, The University of Hong Kong, Pokfulam, Hong Kong, China; pcsham@hku.hk

**Keywords:** genome-wide association studies, SNP heritability, genetic epidemiology, bioinformatics, immunogenetics

## Abstract

Genome-wide association studies (GWAS) are commonly employed to study the genetic basis of complex traits/diseases, and a key question is how much heritability could be explained by all single nucleotide polymorphisms (SNPs) in GWAS. One widely used approach that relies on summary statistics only is linkage disequilibrium score regression (LDSC); however, this approach requires certain assumptions about the effects of SNPs (e.g., all SNPs contribute to heritability and each SNP contributes equal variance). More flexible modeling methods may be useful. We previously developed an approach recovering the “true” effect sizes from a set of observed *z*-statistics with an empirical Bayes approach, using only summary statistics. However, methods for standard error (SE) estimation are not available yet, limiting the interpretation of our results and the applicability of the approach. In this study, we developed several resampling-based approaches to estimate the SE of SNP-based heritability, including two jackknife and three parametric bootstrap methods. The resampling procedures are performed at the SNP level as it is most common to estimate heritability from GWAS summary statistics alone. Simulations showed that the delete-*d*-jackknife and parametric bootstrap approaches provide good estimates of the SE. In particular, the parametric bootstrap approaches yield the lowest root-mean-squared-error (RMSE) of the true SE. We also explored various methods for constructing confidence intervals (CIs). In addition, we applied our method to estimate the SNP-based heritability of 12 immune-related traits (levels of cytokines and growth factors) to shed light on their genetic architecture. We also implemented the methods to compute the sum of heritability explained and the corresponding SE in an R package SumVg. In conclusion, SumVg may provide a useful alternative tool for calculating SNP heritability and estimating SE/CI, which does not rely on distributional assumptions of SNP effects.

## 1. Introduction

Genome-wide association studies (GWAS) have proven to be successful in dissecting the genetic basis of a variety of diseases. A number of new susceptibility loci have been discovered, providing novel insight into the pathophysiology of many diseases. Nevertheless, a large proportion of heritability still remains unexplained. It is natural to question the maximum variance that could be explained by all variants in a GWAS (or meta-analyses of GWAS), as we expect that many true susceptibility variants are “hidden” due to limited power.

A number of methods have been developed to estimate total heritability according to all measured SNPs (also known as SNP-based heritability). Regarding methods that require individual-level data, in a pioneering work, Yang et al. [1] derived a method to estimate the variance explained by all SNPs in a GWAS using a linear mixed model with random SNP effects. The approach assumes that all SNPs have non-zero and normally distributed effects (beta), with a mean effect of zero. Each SNP is assumed to contribute to the same level of explained variance (i.e., variance explained by each SNP = total heritability/number of SNPs). Other similar approaches have also been proposed. For example, LDAK [2] assumes that a different heritability explains each SNP, depending on the minor allele frequencies (MAF), linkage disequilibrium (LD) score and imputation quality of the SNP. Advanced methods have also been developed to estimate SNP-based heritability using summary statistics alone. (Here, summary statistics refer to GWAS results for each SNP, with effect size (beta), standard error of beta and test statistics/p-values available, or at least two items available.) LD score regression (LDSC) is one of the most widely used approaches for this purpose [3]. LDSC assumes a mean effect (beta) of zero and equal variance explained by each SNP (i.e., an infinitesimal model). SumHer [4] is an alternative approach based on the LDAK assumptions. For a more detailed technical review, please refer to ref [5]. The broader problem of SNP-based heritability estimation has also been discussed in several other reviews or opinion pieces [6,7,8,9].

Prior to the development of LDSC, we have developed an alternative framework (see ref [10]; referred to as “SumVg” in this paper) to achieve the same goal of estimating SNP-based heritability using summary statistics alone. Essentially, we aimed to recover the *true* effect sizes from a set of *observed z*-statistics based on formulas presented by Robbins [11] (who attributed the idea to Maurice Kenneth Tweedie), Brown [12] and Efron [13]. The corrected *z*-statistics are then converted to variance explained. There are several advantages to this method. Most importantly, the SumVg approach does *not* rely on any distributional assumptions of the effect sizes of susceptibility variants. In addition, it does *not* assume an equal amount of heritability is explained by each SNP, or that all SNPs contribute to the heritability (infinitesimal model). There are also no assumptions about the relationship between allele frequencies and variance explained. The method is also computationally fast. In addition, since the LDSC method directly leverages LD patterns, a well-matched LD reference panel is usually required [14]. There is less reliance on LD information when using SumVg as LD is mainly used for pruning.

Our method has been applied in a number of studies (for example see [15,16,17,18,19,20,21,22,23]). However, there are no methods available to quantify the standard error (SE) or precision of the heritability estimates from SumVg, or the corresponding confidence intervals (CIs). There is considerable technical difficulty in developing a reliable approach for estimating the SE since usually only the summary data (instead of individual-level data) are available. If raw data are available, a standard non-parametric bootstrap could be employed by sampling individuals with a replacement. However, there are currently no methods for evaluating the SE or CI of the point estimate of heritability when only summary statistics are available.

We summarize the contributions of this study below. In this work, we proposed five re-sampling approaches to estimate the SE of the total heritability of all SNPs in GWAS, based on summary statistics alone. Extensive simulations were performed to compare and validate the performance of different methods. We also explored various methods for constructing CIs. Secondly, we also developed an easy-to-use R program to implement the SumVg approach with different flexible modeling options, available at https://github.com/lab-hcso/Estimating-SE-of-total-heritability/ (accessed on 12 October 2023). Thirdly, we reported heritability estimates for 12 immune-related traits (levels of cytokines and growth factors) [24] based on this approach, for which LDSC was unable to provide reasonable estimates. Such cytokines/growth factors are regulators of immune responses and inflammation, and are important intermediate phenotypes for autoimmune, inflammatory and infectious diseases [25]. As such, it is of scientific and clinical importance to unravel the genetic architecture of these traits, and estimating their heritability may be considered a useful contribution in its own right.

## 2. Results

### 2.1. Overview of Methods

We estimated the total heritability (Vg) explained by all variants in a GWAS panel using the Tweedie’s formula [10], which corrects selection bias in the observed *z*-statistics. To estimate the standard errors (SE), we proposed five resampling methods. The first two are based on jackknife, namely delete-one and delete-*d*-jackknife (with *d* = *n*/5 observations removed each time). We also proposed three parametric bootstrap methods, where *z*-statistics were sampled from a normal distribution based on the ‘corrected’ *z*-statistics, and/or the local false discovery rate (fdr) (i.e., estimated probability that a SNP is null). We also proposed several methods for constructing confidence intervals (CIs), including normal approximation with various bootstrap bias corrections, as well as the percentile and union of CI methods. We tested the performance of the SE and CI estimation methods in simulations under different heritability and sample size scenarios. We applied our methods to estimate SNP-based heritability and the SEs of 12 immune traits to reveal their genetic architecture.

### 2.2. Simulation Results for SE Estimation

Standard errors (SEs) of heritability, as estimated by the jackknife and bootstrap approaches, are listed in Table 1 and plotted in Figure 1. Bias, variance and root mean square error (RMSE) of SEs were calculated over 100 simulations (Table 2; Figure 1 and Figure 2). 

The delete-[*n*/5]-jackknife worked reasonably well when the total heritability explained was low (when heritability = 0.101), but it tended to overestimate the SE when the total heritability was higher, especially with larger sample sizes. The bias was also positive across all simulation scenarios. The standard (delete-1) jackknife approach performed the worst among all methods, producing inflated estimates of SE. The variance and RMSE of this estimator were high compared to other approaches. The SE was, in general, over-estimated at all heritability levels across all sample sizes. This may be explained by the fact that the sum of variance explained is not a very smooth parameter, which impairs the performance of delete-1-jackknife estimators.

The other methods, including the original parametric bootstrap (paraboot) and the modified versions with consideration of local fdr, performed reasonably well and closely resembled the true SE. With the exception of one simulation setting, the parametric bootstrap methods achieved the lowest (absolute) bias for SE. For the variance and RMSE of SE, parametric bootstrap also performed the best. In terms of RMSE, the parametric bootstrap approaches modeling the local fdr (i.e., fdrboot1 and fdrboot2) outperformed the other methods. The RMSE of different estimators were also observed to reduce with increasing sample sizes.

### 2.3. Performance of Different CI Construction Methods

The full results are presented in Table 3, Appendix A. For standard CI (based on normal approximation), the CIs built from the SE of delete-*d*-jackknife performed reasonably well (in terms of coverage) for large sample sizes, although the coverage was not always adequate for modest samples sizes, especially for *N* < 20,000. The coverage of CIs constructed from other types of SEs were more variable, with good coverage for some scenarios but poor coverage for others. Therefore, we primarily focus on the SE from delete-*d*-jackknife when a standard CI is used. Interestingly, the bias-corrected standard CI, with bias correction based on paraboot or fdrboot2, performed better in the several cases when the standard CI had low coverage (<50%) (we assume that the SE from delete-*d*-jackknife was employed). The performance of percentile CIs was highly variable across different scenarios.

In view of the highly variable performance of different CI construction methods, we expect the union of CI (UCI) to perform better and be more robust across different scenarios. We observed that UCI, no matter if it is constructed from the standard or percentile CI estimators, in general, achieved good coverage across most simulation scenarios, although in some cases the coverage was still below the desired level (95%). When we further took the union of standard and percentile UCI estimators (i.e., Method 3 listed under ‘Union CI’ in the Section 4), the coverage was adequate for almost all scenarios, except one case in which both the sample size and the sum of variance explained (Vg) were low (*N* = 5000, Vg = 0.101).

### 2.4. Results on Immune Traits

PLINK was applied to trim GWAS data for 12 immunological traits (Table 4) with various *r^2^* criteria to obtain roughly independent SNPs. We only included common variants with an MAF > 0.01 for further analysis. Then, using SumVg, the “true” *z*-statistics of trimmed SNPs were retrieved to capture the missing heritability. The jackknife and bootstrap methods were used to compute the corresponding SEs (Table 5; Appendix A).

The total SNP-based heritability predicted by SumVg for the selected traits, in contrast to the comparatively low or negative heritability estimates from LDSC, were around 10–20% based on a collection of LD-pruned SNPs. We obtained a stable (and likely conservative) estimate of heritability at *r*^2^ ~ 0.01 or 0.005. Lower *r*^2^ values (i.e., *r*^2^ < 0.0025 and *r*^2^ < 0.001) had limited impact on final estimates of heritability. The delete-one jackknife consistently produced the highest standard error, while the bootstrap and delete-*d* jackknife approaches produced SEs that were more comparable to one another. Out of the 12 cytokines/growth factors studied, the highest heritability was observed for the levels of IL-4 and IL-17.

### 2.5. R Package Implementation

We also implemented the methods to compute the sum of heritability explained and the corresponding SEs in an R package SumVg, available at https://github.com/lab-hcso/Estimating-SE-of-total-heritability/ (accessed on 12 October 2023).

The computational speed of different resampling approaches using SumVg is presented in Appendix A (assuming 100,000 SNPs and 200 resampling iterations). The speed is generally fast and the time taken was around 2–4 min for each resampling method, using a single core (Intel Xeon Gold 6230 CPU @ 2.10 GHz).

## 3. Discussion

In this study, we presented an approach for estimating the SE of SNP-based heritability estimates using SumVg, and our applications to immune phenotypes demonstrate the usefulness of this approach.

Our main purpose is to provide an alternative approach for SNP-based heritability and SE estimation, since different approaches have different statistical modeling assumptions, or assumptions about the genetic architecture. In practice, it is almost impossible to know the true genetic architecture of a disease/trait, and as such, it is very difficult to verify the correctness of heritability estimates due to the lack of a ‘gold standard’. It will be more reassuring if one observes similar heritability estimates from diverse methods. SumVg may provide a useful alternative reference for heritability estimates, in conjunction with existing approaches such as LDSC. SumVg may also be useful when standard approaches are unable to give reasonable results (e.g., close to zero heritability for traits that are likely to be heritable from previous studies, or negative estimates). It will be interesting to investigate the reasons underlying negative heritability estimates for LDSC; one possibility is mis-specified model assumptions [26], but the exact reasons will require further studies.

We recommended pruning the SNPs (such that SNPs are roughly in linkage equilibrium) before applying our method of heritability estimation. One approach is to employ a series of *r*^2^ thresholds (e.g., decreasing *r*^2^ from 0.1 to 0.001) and consider the point at which heritability became stable. Our empirical applications showed that an *r*^2^ threshold of ~0.01 may be sufficient. The resulting SNP-based heritability may be considered to be a conservative estimate (due to the possibility of removing some causal variants during LD-pruning). While not directly modeling LD is a limitation of this approach, the lower reliance on accurate LD information may be advantageous in some cases, for example when in-sample LD information is not available and only limited external reference data are present. On the other hand, we are also investigating methods to model LD in the SumVg framework. Since SumVg and LDSC are based on different modeling strategies and assumptions, and that the main focus of this study is the development of new SE/CI estimation approaches for SumVg (as well as applications to immune traits and presentation of a new R package), we shall leave carrying out a detailed comparison between SumVg and LDSC (or other SNP heritability estimation methods) for future work.

We have not investigated methods for SE estimation when raw genotype data are available. When raw data is available, one potential approach is to simply resample the individuals with a replacement (i.e., standard non-parametric bootstrap). However, such an approach is computationally intensive and its performance over methods based on summary statistics requires further research. The above resampling methods can also potentially be sped up by splitting the job into multiple processes to be run in parallel, although this approach has not been implemented in our software yet. We also wish to point out that, as the resampling methods were supposed to apply to GWAS summary data, in general the computational speed is fast, and the speed is not affected by sample sizes.

We have explored various approaches to construct CI, although we cannot yet find a single approach that yields an optimal CI with good coverage across all scenarios. We shall leave the development of more sophisticated and novel methodologies for CI construction for future works. For practical purposes, the union CI appears to perform well in terms of coverage across most scenarios (at the expense of wider CIs). On the other hand, we suspect that the issue of CI construction may not be unique to the SumVg approach; other methods for estimating SNP-based heritability typically require more stringent assumption on the distribution of effects, and/or that all SNPs contribute to heritability. The violation of such assumptions may lead to the estimates being biased and the inadequate coverage of CIs. Here we have proposed a bootstrap correction of bias, which indeed led to improvement in CI coverage in some cases, for example the standard CI under small sample sizes. Nevertheless, bootstrap correction showed a variable performance across different scenarios and did not always reduce bias in all cases. The above issues may warrant further studies.

Here we further highlight several important points to note and limitations of our framework. Regarding the SumVg estimator of total SNP-based heritability, one future research direction is to further explore its asymptotic theoretical properties. We did not pursue this direction here. Of note, the key difficulty in Equation (1) (i.e., the Tweedie’s formula) is to estimate $f\left(x\right)$ and $f^{\prime }\left(x\right)$ accurately. We primarily employed a kernel density estimator here, although other density estimation approaches may also be attempted. Notably, the kernel density estimator has been shown to be asymptotically consistent under certain assumptions [27]. In the paper by Efron [28], the asymptotic regret (Reg) of the empirical Bayes approach (i.e, using Tweedie’s formula) was studied by comparing the Tweedie’s estimate with the Bayes estimate of the true effect size, for a fixed value of z at $z_{0}$. It was shown that Reg$\left(z_{0}\right)$ tends towards zero as *N* tends towards infinity, and the regret depends on the squared error of $\hat{l^{\prime }}\left(z_{0}\right)$ as an estimator of $l^{\prime }\left(z_{0}\right)$, where $l^{\prime }\left(z\right)=\frac{d}{dz}\;\log\;f\left(z\right)$. Future theoretical studies of SumVg and other SNP heritability estimation methods are warranted.

In the current work, we assume that the summary statistics have been corrected for population stratification and other types of bias. If the original GWAS study suffered from bias, e.g., confounding, selection/ascertainment bias, sampling bias, bias due to missing data, etc., the resulting Vg estimate will also be affected. We suggest that the above bias should be carefully addressed at the design and/or analysis stage of the GWAS, for example by performing proper random sampling, inverse probability weighting to address selection bias [29], proper imputation of missing data, etc. As with any method, independent replication is also important.

Another limitation is that the proposed approach for calculating SNP heritability and SE/CI estimation may not work well for very small sample sizes. Since GWAS sample sizes are generally getting larger (most with *N* > 5000), we did not address the performance under very small sample sizes here. In such cases, both the SNP heritability and SE estimates may need to be viewed with greater caution. Meta-analysis of GWAS results across multiple studies may be recommended. Future work may also explore more innovative approaches to addressing small sample sizes, for example whether specifying a prior for the underlying effect sizes (*δ*) may help. (The current approach does not require any specification of the distribution of *δ*).

We also note that resampling methods often assume that the data points are independent of each other. In our study, prior to the analysis, we processed the data to remove strongly linked SNPs using LD pruning. The resulting SNPs are therefore roughly independent though some residual LD might remain. As a future direction, it may be useful to explore ways to fully tackle LD, for example by block bootstrap or jackknife [30]. However, external LD data from reference panels would be required, and there may be risks of LD mismatch between the studied and external samples. Further studies are required to investigate these issues.

Different resampling methods like bootstrap and jackknife may have different assumptions and applicability to different kinds of data. We have conducted relatively extensive simulations to compare performance of different methods across a range of heritability levels and sample sizes, which helps evaluate their applicability. We believe the proposed methods are generally applicable to most GWAS summary data. Note that the parametric bootstrap approaches assume that the observed data (*z*-statistics) are drawn from a certain specified parametric distribution. In our case, it is assumed that the *δ* and/or local fdr are estimated reasonably well. For small sample sizes, this assumption may not hold very well. The jackknife approaches do not require parametric assumptions; however, delete-one-jackknife has been shown to produce inconsistent variance estimators for non-smooth estimators such as the sample quantiles [31]. Delete-*d*-jackknife can resolve this problem, but the choice of *d* may not be straightforward. We suggest that multiple types of resampling methods should be performed; similar results across different methods may provide reassurance to the validity of results. Future work may include more extensive simulations for different genetic architectures and wider applications to complex traits.

There may be a concern that resampling methods may not handle extreme values or skewed distributions well. As discussed above, we recommend the GWAS should be conducted carefully in the first place. For example, skewed phenotypes may require transformation before analysis, and confounding or other kinds of bias need to be addressed. The SumVg method works on summary statistics. It is possible to perform further inverse-rank transformation to the summary statistics if the distribution is skewed or outliers are present, although this may create some bias to the Vg estimate. One may also trim the outlying *z*-statistics, and increasing the number of resamples may also help. The performance of these approaches will be a topic for further studies.

Importantly, we have also applied our approach to estimate the heritability of different cytokines, which play important roles in immune response and the pathogenesis of autoimmune, inflammatory and infectious diseases. Our analyses suggest that the studied cytokines are moderately heritable in general.

To summarize, SumVg is useful for triangulating evidence from different approaches to support conclusions regarding SNP-based heritability. We present novel methods of computing SE and CI and an easy-to-use software here, which we believe will be helpful for other researchers. Our application to the cytokine levels also sheds light on the genetic architecture of these clinically important immune traits.

## 4. Materials and Methods

### 4.1. Estimation of the Total Heritability Explained (Vg)

We previously proposed an approach [10] to estimate the sum of heritability explained by all variants on a GWAS panel. Our approach leverages Tweedie’s formula for estimating the true underlying effect sizes of SNPs, based on the observed GWAS summary statistics. The principles are described in detail in the work by Efron [28].

#### 4.1.1. Estimation of Total Vg Based on Tweedie’s Formula

More specifically, assuming we have a large number of normally distributed variables (here *z*-statistics from a GWAS analysis), each with its own unobserved mean parameter *δ_i_*, then
$$z_{i}\sim N\left(\delta_{i},\;\sigma^{2}\right)\;\;,\;\;\;i=1,\;2,\;\ldots ,\;k$$
where *k* is the total number of variables. The attention is focused on the more extreme values, for example the top SNPs in high-dimensional genomics studies. As described by Efron [28], ‘selection bias’ may be at play here. Intuitively, the more extreme *z*-statistics might have been ‘lucky’ as random errors pushed them to deviate from zero; as such they can ‘stand out’ among the other *z*-statistics. In other words, the true underlying effect sizes of these top SNPs tend to be less extreme than the observed values. This phenomenon is also known as the ‘winner’s curse’, for example see [32,33,34]. As a result, if we directly used the observed *z*-statistics to estimate the true effect sizes, the performance may not be optimal. Some form of ‘correction’ of the observed *z*-statistics are required.

Efron [28] proposed an empirical Bayes approach to reduce the selection bias, which was first described by Robbins [11] who attributed the ideas to Tweedie. The method assumes that
$$\delta \;\sim \;g\left(\;\cdot \;\right)\;\;\;\;\;\;and\;\;\;z|\delta \;∼N\left(\delta ,\;\sigma^{2}\right)$$

In other words, we assume that *δ* was sampled from a prior ‘density’ *g*(.), then *z* ~ 𝒩(*δ*, *σ*^2^) were observed, and the variance *σ*^2^ was known. There are no assumptions on the form of the prior density *g*. According to the Tweedie’s formula,
$$E\{\delta \;|\;z\}=z+\sigma^{2}l\prime (z),\mathrm{where}\;l\prime (z)=\frac{d}{dz}log\;f(z)$$

In our setting of GWAS analyses, we assume *σ*^2^ = 1, since we work with the summary *z*-statistics. We estimated the true or ‘corrected’ effect sizes of SNPs using
(1)$$E\left\{\delta |z\right\}=\;z+\frac{f^{\prime }\left(z\right)}{f\left(z\right)}$$
which is equivalent to the formula above when *σ*^2^ = 1. Here *z* denotes the observed *z*-statistic, obtained from the estimated regression coefficient divided by the estimated SE (i.e., $\frac{\hat{\beta }}{\hat{SE}}$). *δ* is the *z*-statistic derived by the *true* effect size divided by the estimated SE of the sample $(\beta_{true}/\hat{SE}$), which can be considered a form of the ‘standardized’ true effect size. We previously proposed to employ a kernel density estimator to compute *f*(*z*) [10], which was shown to perform well in simulations. The total variance explained (Vg) can be obtained by converting the underlying effects *δ* to the Vg scale (see below and ref [10]).

#### 4.1.2. Conversion of *z*-Statistics to Vg

For continuous traits, the conversion formula followed our previous work [10], which can be derived from ANOVA table of regression,
$$Vg=\frac{{\left[E\left(\delta |z\right)\right]}^{2}}{n-2+{\left[E\left(\delta |z\right)\right]}^{2}}$$

For binary outcomes, it is also possible to convert the *z*-statistics to Vg, provided that the estimated SE (or beta) and minor allele frequencies (MAF) of the SNPs, as well as the outcome prevalence, are available. We followed the methodology described in ref [35], which described how to convert coefficients from a logistic model to the liability scale. Note that the liability is assumed to have a variance of one. We followed Equation (4) from the above paper [35] to derive the coefficient (*τ*_1_) under a liability scale. We converted *τ*_1_ to the standardized coefficient (*τ_standard_*) by multiplying *τ*_1_ by sqrt(2 × MAF × (1 − MAF)), which is the standard deviation (SD) of the allelic count (coded as 0, 1, 2). Total variance explained is given by sum of the squared *τ_standard_*.

#### 4.1.3. Assumptions

Regarding the assumptions of this approach, we emphasize that it does *not* require prior assumptions about the underlying distributions of the true effect sizes δ, which is an important advantage over other SNP-heritability estimation methods. On the other hand, we assume that the summary statistics have been corrected for population stratification or other confounding factors. The *z*-statistics are assumed to follow normal distributions; for very small samples sizes, rare variants, highly imbalanced case to control ratio, or highly skewed continuous outcomes, etc., caution should be taken as to whether the test statistic $\frac{\hat{\beta }}{\hat{SE}}\;$ follows a normal distribution. We assume full GWAS summary statistics as input; if the summary statistics have been selected based on their significance levels (e.g., some GWAS only released the top SNPs, say top 10,000 SNPs), the proposed Tweedie’s formula may not work well. The effect sizes may be overestimated in this case as the other SNPs have been selected for being significant.

#### 4.1.4. An Alternative Conditional Estimator

We also proposed an alternative approach by evaluating the expected effect size conditioned on H1 (i.e., δ≠ 0)
(2)$$E\left(\delta |z,\;H_{1}\right)=E_{1}\left(\delta |z\right)=\frac{E\left(\delta |z\right)}{Pr\left(H_{1}|z\right)}=\frac{E\left(\delta |z\right)}{1-fdr\left(z\right)}$$
where fdr is the local false discovery rate described in Efron [36]. The resulting estimate of Vg can be obtained by first converting $E\left(\delta |z,\;H_{1}\right)$ to the Vg scale (see Section 4.1.2), then multiply by 1 − *fdr*(*z*).

The conditional estimator, however, is prone to large random variations as it involves local fdr estimation of each SNP. In many subsequent applications of our heritability estimation method [15,16,17], the unconditional estimator (Equation (1)) was primarily employed. We shall hence focus on the unconditional estimator in this paper, although the resampling approaches described below can readily be applied to other estimators in our previous work [10] as well.

### 4.2. Estimation of the Standard Error (SE) of Vg

#### 4.2.1. Standard and Delete-*d*-Jackknife to Estimate SE

In standard (delete-one) jackknife procedure [37], we estimate the standard error (SE) by leaving out one observation at a time. The SE is defined by
$${\hat{se}}_{jack}=\sqrt{\frac{n-1}{n}\sum {\left({\hat{\theta }}_{\left(i\right)}-{\hat{\theta }}_{\left(.\right)}\right)}^{2}}$$
where *n* is the sample size, ${\hat{\theta }}_{\left(i\right)}$ is the parameter estimate from the sample with the *i*th observation removed and
$${\hat{\theta }}_{\left(.\right)}=\sum_{i=1}^{n}\frac{{\hat{\theta }}_{\left(i\right)}}{n}$$

In our case, the parameter is the sum of heritability from all variants.

An extension is the delete-*d*-jackknife [31] where we leave out *d* observations at a time. There are in total $N=\left(\begin{array}{c} n\\ d \end{array}\right)$ possibilities of removing *d* out of *n* observations. In practice, *N* is usually very large. One may simply randomly repeat the procedure *m* times only $\left(m\leq N\right)$ instead of exhausting all possibilities of removing *d* out of *n* observations. The standard error is given by
$${\hat{se}}_{del-d-jack}=\sqrt{\frac{n-d}{dm}\sum_{v=1}^{m}{\left({\hat{\theta }}_{S_{v}}-\frac{1}{m}\sum_{v=1}^{m}{\hat{\theta }}_{S_{v}}\right)}^{2}}$$
where ${\hat{\theta }}_{S_{v}}$ denotes the parameter estimate in the *v*th jackknife replicate where *d* observations are left out. The delete-*d*-jackknife (when d>1) works better than the standard jackknife for non-smooth parameters like the median [31].

There are no clear rules on the choice of *d* in delete-*d*-bootstrap. Chatterjee [38] suggested *n*/5 as a reasonable choice for *d* based on the consideration of efficiency and likely model conditions. We followed the suggestion by Chatterjee [38] and set *d* as *n*/5 (=20,000) in all simulations.

#### 4.2.2. Parametric Bootstrap Approaches for Estimating SE

In parametric bootstrap, in each replication we simulated *z*-statistics based on $\hat{\delta }$, the ‘corrected’ *z*-statistics from original sample (this method is referred to as ‘paraboot’). We have
zi,b~Nδi^,1
where $z_{i,b}$ denotes the *i*th *z*-statistic in the *b*th bootstrap replicate. For small effects, the $\hat{\delta }$ will be shrunken towards zero.

We further proposed a modified approach by also considering the local fdr (i.e., probably of null given *z*) of each *z*-statistic. In each replicate, we simulated *z*-statistics according to the following scheme:zi,b~Nzi^,1 with a probability of 1-fd^r(zi)
$$z_{i,b}\sim N\left(0,1\right)\;\mathrm{with}\;a\;\mathrm{probability}\;\mathrm{of}\;f\hat{d}r(z_{i})$$
where zi^ denotes the observed *z*-statistics. The standard error is then computed from the simulated *z*-statistics. This method is referred to as “fdrboot1”.

Alternatively, one may employ the corrected *z*-statistics instead of the observed *z*-statistics as the mean in each simulation, i.e.,
zi,b~Nδi^,1 with a probability of 1-fd^r(zi)
$$z_{i,b}\sim N\left(0,1\right)\;\mathrm{with}\;a\;\mathrm{probability}\;\mathrm{of}\;f\hat{d}r(z_{i})$$

The method is also referred to as ‘fdrboot2’.

### 4.3. Construction of Confidence Intervals (CIs): An Exploratory Analysis

The construction of a proper CI is a more demanding task as it requires the unbiasedness of the estimate and correct estimation of the variability of the estimate. Given the difficulty of constructing accurate CIs, here we consider CI estimation as a secondary or exploratory analysis which requires further investigation and methodological development. We have explored a few approaches as described below.

#### 4.3.1. Normal Approximation (Standard Approach)

Firstly, we explored the standard approach for constructing the 95% CI by using normal approximation, i.e., $\hat{Vg}\pm \;z_{0.975}\times \hat{SE}\left(Vg\right)$, where $\;z_{0.975}$ is the quantile of a standard normal distribution at the 97.5th percentile. Assuming a polygenic model, the total heritability is the sum of variance explained contributed by many variants of small to modest effect sizes. Hence, it is reasonable to assume normality according to the central limit theorem (as is assumed by other SNP-heritability estimation tools). We examined the performance of different CIs, with SE determined by various methods. Empirically, we found that SE computed by the delete-*d*-jackknife performed reasonably well.

On the other hand, we also explored this using bootstrap to correct for bias of the point estimates of Vg. In brief, the bias can be estimated by [39]
$$\hat{Bias}=\bar{\theta^{*}}-\hat{\theta }$$
where $\hat{\theta }$ denotes the observed Vg, and $\bar{\theta^{*}}$ is the mean of the bootstrapped estimates of Vg. The bias-corrected estimator of Vg is given by
$$Bias\;corrected\;\theta =\hat{\theta }-\hat{Bias}$$

The 95% CI is then based on $\hat{Bias\;corrected\;Vg\pm z_{0.975}\times \hat{SE}}$. Since we proposed 3 bootstrap procedures, there were 3 bootstrap bias-corrected CIs based on normal approximation. The standard CI without bias correction was also included as another estimator.

#### 4.3.2. Percentile Approach

Secondly, we explored the percentile CI approach, namely construction of 95% CIs based on the 2.5th and 97.5th percentiles of the bootstrapped Vg. Again, bias correction can be applied as follows
$$\mathrm{Lower}\;95\%\;\mathrm{CI}=2\hat{\theta }-\theta_{\left(0.975\right)}^{*}$$
$$Upper\;95\%\;CI=2\hat{\theta }-\theta_{\left(0.025\right)}^{*}$$
where $\theta_{\left(0.025\right)}^{*}$ and $\theta_{\left(0.975\right)}^{*}$ are the 2.5th and 97.5th percentiles of the bootstrapped replicates of Vg, respectively. Bias correction was based on the same bootstrap method that was used to derive the percentiles. Again, we also included the percentile CIs without bias correction.

#### 4.3.3. Union CI

Thirdly, we explored a more robust CI estimator by taking the union of individual CIs (UCI). The union of multiple CIs is constructed by taking the minimum of the lower CIs across different methods as the final lower CI, and the maximum of different upper CIs as the final upper CI. This union approach can ensure better robustness if CI construction approaches perform differently under different scenarios. The UCI method has been widely employed in instrumental variables regression to improve robustness of results in the presence of pleiotropy [40].

In summary, the following methods were explored:Normal approximation (standard approach), without bias correction (one estimator) or with bootstrap bias correction (3 estimators), then take the union of CIs;Percentile approach, without bias correction (3 estimators) and with bias correction (3 estimators), then take the union of CIs;Union of the final CI obtained from 1 and 2.

### 4.4. Simulation Studies

We compare the SE estimated from the above methods with the ‘true’ SE obtained from one hundred simulations with known data generating distributions. The details of the simulations is as follows [10]. Briefly, a gamma distribution was used to simulate three levels of variance explained (Vg = 0.101, 0.191, 0.295), which were converted to true effect sizes (*δ*). *Z*-statistics for 100,000 independent SNPs (0.5% were non-null) with different sample sizes (*N* = 5000, 10,000, 20,000, 50,000, 100,000, 200,000) were then simulated as input for SumVg following the distribution *N*(*δ*,1). Two hundred replicates were run for each bootstrap or jackknife procedure. We focus on quantitative traits in our simulations, but the results should most likely apply to binary traits as well, as the only difference in these two scenarios is the formula to convert *z* to variance explained (Vg). The performance of different methods for CI construction was also evaluated.

### 4.5. Application to Immune Traits

A selected set of immune-related traits (levels of cytokines/growth factors) were included for study, based on the GWAS by Ahola-Olli et al. [24]. We selected 12 continuous immune traits with (1) sample size *N* > 5000 and (2) very low (≤3%) or negative SNP-based heritability estimated by LDSC. The LDSC heritability were based on pre-calculated values from GWASAtlas (https://atlas.ctglab.nl/; accessed 1 May 2023). SNPs in strong LD were removed using the PLINK command “--indep-pairwise 100 25 r2” with a series of *r^2^* thresholds (0.1, 0.05, 0.025, 0.01, 0.005, 0.002, 0.001). The 1000G Phase3 EUR sample was used as the reference panel to calculate LD among variants. Independent SNPs with MAF > 0.01 were then applied to SumVg.

## Figures and Tables

**Figure 1 ijms-25-01347-f001:**
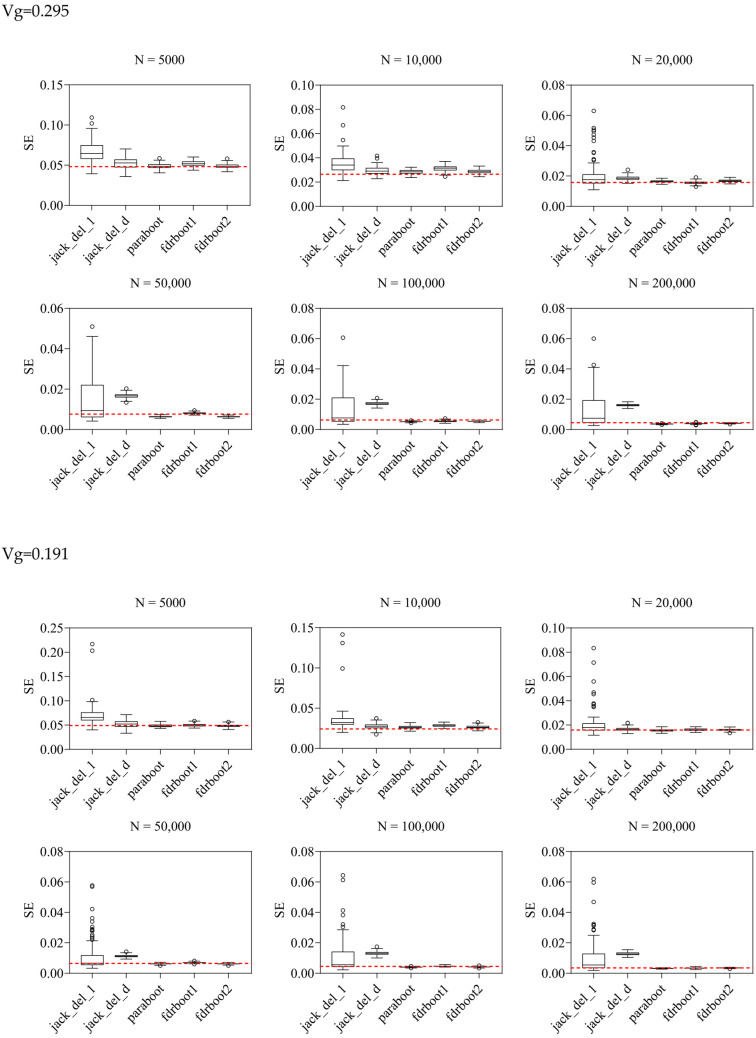
Boxplots of SE estimated by different approaches. Vg is the sum of variance explained, *N* is the sample size, and the horizontal line refers to true SE calculated by repeating the experiments 100 times based on the true data generating mechanism. jack_del_1, jack_del_d, paraboot, fdrboot1 and fdrboot2 are different SE estimation approaches as described above.

**Figure 2 ijms-25-01347-f002:**
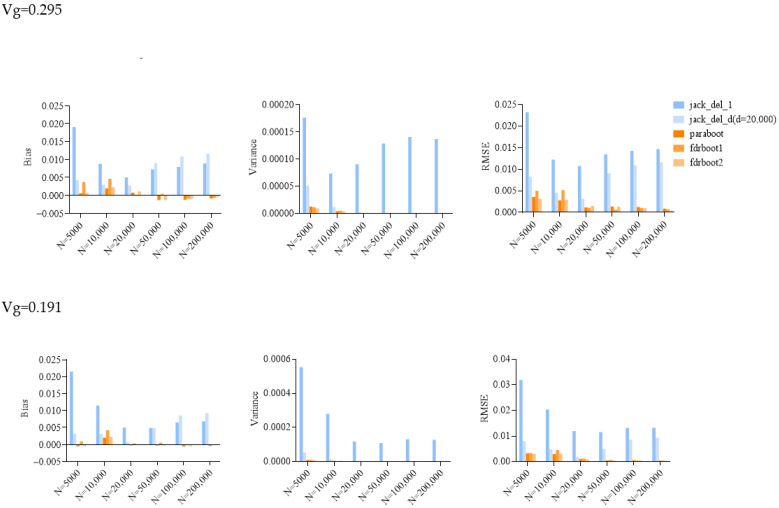
Bar plots of bias, variance and root mean squared error (RMSE) of SE estimated by different approaches in simulations. Vg is the sum of variance explained, and N is the sample size. jack_del_1, jack_del_d, paraboot, fdrboot1 and fdrboot2 are different SE estimation approaches as described above.

**Table 1 ijms-25-01347-t001:** Standard error (SE) of the sum of variance explained (Vg) estimated by different resampling approaches.

Sum_of_Vg	Sample_Size	Mean_Est	TRUE_SE	SE				
				jack_del_1	jack_del_d	paraboot	fdrboot1	fdrboot2
0.295	5000	0.232	0.0482	0.0672	0.0524	0.0488	0.0519	0.0489
	10,000	0.210	0.0265	0.0353	0.0295	0.0285	0.0312	0.0287
	20,000	0.244	0.0158	0.0208	0.0185	0.0165	0.0156	0.0168
	50,000	0.283	0.0076	0.0149	0.0167	0.0063	0.0081	0.0063
	$1\times 10^{5}$	0.312	0.0063	0.0143	0.0172	0.0051	0.0055	0.0054
	$2\times 10^{5}$	0.321	0.0045	0.0134	0.0161	0.0036	0.0038	0.0041
0.191	5000	0.207	0.0491	0.0706	0.0523	0.0486	0.0500	0.0485
	10,000	0.147	0.0242	0.0357	0.0274	0.0263	0.0285	0.0265
	20,000	0.158	0.0159	0.0208	0.0166	0.0156	0.0162	0.0160
	50,000	0.174	0.0064	0.0113	0.0113	0.0061	0.0070	0.0061
	$1\times 10^{5}$	0.195	0.0045	0.0110	0.0131	0.0040	0.0047	0.0041
	$2\times 10^{5}$	0.207	0.0035	0.0103	0.0128	0.0031	0.0034	0.0035
0.101	5000	0.197	0.0521	0.0692	0.0524	0.0484	0.0496	0.0483
	10,000	0.116	0.0260	0.0345	0.0265	0.0251	0.0257	0.0251
	20,000	0.098	0.0143	0.0202	0.0159	0.0150	0.0153	0.0154
	50,000	0.091	0.0058	0.0098	0.0078	0.0063	0.0057	0.0063
	$1\times 10^{5}$	0.094	0.0032	0.0069	0.0076	0.0027	0.0036	0.0027
	$2\times 10^{5}$	0.107	0.0028	0.0072	0.0083	0.0023	0.0027	0.0025

The main purpose of this table is to compare the true SEs against the SEs estimated by various resampling-based methods. Sum_of_Vg, true total heritability explained (i.e., the real total heritability based on our data-generating mechanism); Sample_size refers to the sample size of the GWAS; Mean_Est, the mean *estimated* heritability explained based on our approach of corrected *z*-statistics; True_SE, the ‘true’ SE was based on repeating our simulation experiments 100 times; jack_del_1, delete-1-jackknife; jack_del_d, delete-*d*-jackknife with d equal to 20,000; paraboot, parametric bootstrap approach as described in the text, based on simulating from a normal distribution in which the mean was derived from the corrected *z*-statistics (without consideration of local false discovery rates (fdr)); fdrboot1, a “weighted” bootstrap approach with consideration of the local fdr, using the observed *z*-statistic as the mean in each simulation; fdrboot2, a “weighted” bootstrap approach with consideration of the local fdr, using the corrected *z*-statistic as the mean in each simulation.

**Table 2 ijms-25-01347-t002:** Bias, variance and root mean squared error (RMSE) of SE estimated by different resampling approaches.

Sum_Vg	*N*	Bias of the Estimator for SE					Variance of the Estimator for SE					RMSE of the Estimator for SE				
		jack_del_1	jack_del_d	paraboot	fdrboot1	fdrboot2	jack_del_1	jack_del_d	paraboot	fdrboot1	fdrboot2	jack_del_1	jack_del_d	paraboot	fdrboot1	fdrboot2
0.295	5000	1.91 $\times \;10^{-2}$	4.26 $\times \;10^{-3}$	**5.92 × 10^−4^**	3.73 $\times \;10^{-3}$	7.16 $\times \;10^{-4}$	1.77 $\times \;10^{-4}$	5.14 $\times \;10^{-5}$	1.26 $\times \;10^{-5}$	1.15 $\times \;10^{-5}$	**9.30 × 10^−6^**	2.32 $\times \;10^{-2}$	8.34 $\times \;10^{-3}$	3.59 $\times \;10^{-3}$	5.04 $\times \;10^{-3}$	**3.13 × 10^−3^**
	10,000	8.81 $\times \;10^{-3}$	2.98 $\times \;10^{-3}$	**2.00 × 10^−3^**	4.66 $\times \;10^{-3}$	2.25 $\times \;10^{-3}$	7.34 $\times \;10^{-5}$	1.21 $\times \;10^{-5}$	3.87 $\times \;10^{-6}$	4.99 $\times \;10^{-6}$	**3.62 × 10^−6^**	1.23 $\times \;10^{-2}$	4.58 $\times \;10^{-3}$	**2.80 × 10^−3^**	5.17 $\times \;10^{-3}$	2.94 $\times \;10^{-3}$
	20,000	5.04 $\times \;10^{-3}$	2.78 $\times \;10^{-3}$	7.37 $\times \;10^{-4}$	**−1.46 × 10^−4^**	1.07 $\times \;10^{-3}$	9.06 $\times \;10^{-5}$	2.27 $\times \;10^{-6}$	**8.33 × 10^−7^**	1.12 $\times \;10^{-6}$	1.00 $\times \;10^{-6}$	1.08 $\times \;10^{-2}$	3.16 $\times \;10^{-3}$	1.17 $\times \;10^{-3}$	**1.07 × 10^−3^**	1.47 $\times \;10^{-3}$
	50,000	7.25 $\times \;10^{-3}$	9.03 $\times \;10^{-3}$	−1.32 $\times \;10^{-3}$	**4.68 × 10^−4^**	−1.30 $\times \;10^{-3}$	1.29 $\times \;10^{-4}$	1.45 $\times \;10^{-6}$	1.36 $\times \;10^{-7}$	1.93 $\times \;10^{-7}$	**1.30 × 10^−7^**	1.35 $\times \;10^{-2}$	9.11 $\times \;10^{-3}$	1.37 $\times \;10^{-3}$	**6.42 × 10^−4^**	1.35 $\times \;10^{-3}$
	$1\times 10^{5}$	7.97 $\times \;10^{-3}$	1.09 $\times \;10^{-2}$	−1.20 $\times \;10^{-3}$	**−8.78 × 10^−4^**	−9.34 $\times \;10^{-4}$	1.41 $\times \;10^{-4}$	1.52 $\times \;10^{-6}$	**8.11 × 10^−8^**	3.48 $\times \;10^{-7}$	1.00 $\times \;10^{-7}$	1.43 $\times \;10^{-2}$	1.10 $\times \;10^{-2}$	1.23 $\times \;10^{-3}$	1.06 $\times \;10^{-3}$	**9.86 × 10^−4^**
	$2\times 10^{5}$	8.92 $\times \;10^{-3}$	1.16 $\times \;10^{-2}$	−8.57 $\times \;10^{-4}$	−6.32 $\times \;10^{-4}$	**−3.72 × 10^−4^**	1.37 $\times \;10^{-4}$	8.70 $\times \;10^{-7}$	**3.49 × 10^−8^**	1.30 $\times \;10^{-7}$	4.06 $\times \;10^{-8}$	1.47 $\times \;10^{-2}$	1.16 $\times \;10^{-2}$	8.77 $\times \;10^{-4}$	7.27 $\times \;10^{-4}$	**4.23 × 10^−4^**
0.191	5000	2.16 $\times \;10^{-2}$	3.21 $\times \;10^{-3}$	**−5.02 × 10^−4^**	9.69 $\times \;10^{-4}$	−5.23 $\times \;10^{-4}$	5.53 $\times \;10^{-4}$	5.41 $\times \;10^{-5}$	1.07 $\times \;10^{-5}$	1.02 $\times \;10^{-5}$	**8.58 × 10^−6^**	3.19 $\times \;10^{-2}$	8.03 $\times \;10^{-3}$	3.31 $\times \;10^{-3}$	3.34 $\times \;10^{-3}$	**2.98 × 10^−3^**
	10,000	1.15 $\times \;10^{-2}$	3.16 $\times \;10^{-3}$	**2.04 × 10^−3^**	4.22 $\times \;10^{-3}$	2.29 $\times \;10^{-3}$	2.80 $\times \;10^{-4}$	1.43 $\times \;10^{-5}$	4.88 $\times \;10^{-6}$	**2.98 × 10^−6^**	5.03 $\times \;10^{-6}$	2.03 $\times \;10^{-2}$	4.93 $\times \;10^{-3}$	**3.01 × 10^−3^**	4.56 $\times \;10^{-3}$	3.20 $\times \;10^{-3}$
	20,000	4.96 $\times \;10^{-3}$	7.22 $\times \;10^{-4}$	−2.41 $\times \;10^{-4}$	3.15 $\times \;10^{-4}$	**9.85 × 10^−5^**	1.19 $\times \;10^{-4}$	2.70 $\times \;10^{-6}$	1.09 $\times \;10^{-6}$	1.31 $\times \;10^{-6}$	**7.64 × 10^−7^**	1.20 $\times \;10^{-2}$	1.79 $\times \;10^{-3}$	1.07 $\times \;10^{-3}$	1.19 $\times \;10^{-3}$	**8.80 × 10^−4^**
	50,000	4.90 $\times \;10^{-3}$	4.83 $\times \;10^{-3}$	**−2.92 × 10^−4^**	5.76 $\times \;10^{-4}$	−2.97 $\times \;10^{-4}$	1.10 $\times \;10^{-4}$	8.17 $\times \;10^{-7}$	1.66 $\times \;10^{-7}$	1.50 $\times \;10^{-7}$	**1.24 × 10^−7^**	1.16 $\times \;10^{-2}$	4.91 $\times \;10^{-3}$	5.01 $\times \;10^{-4}$	6.94 $\times \;10^{-4}$	**4.60 × 10^−4^**
	${1\times 10}^{5}$	6.45 $\times \;10^{-3}$	8.56 $\times \;10^{-3}$	−5.40 $\times \;10^{-4}$	**1.30 × 10^−4^**	−4.33 $\times \;10^{-4}$	1.32 $\times \;10^{-4}$	1.68 $\times \;10^{-6}$	**6.30 × 10^−8^**	1.32 $\times \;10^{-7}$	8.37 $\times \;10^{-8}$	1.32 $\times \;10^{-2}$	8.65 $\times \;10^{-3}$	5.96 $\times \;10^{-4}$	**3.85 × 10^−4^**	5.21 $\times \;10^{-4}$
	${2\times 10}^{5}$	6.85 $\times \;10^{-3}$	9.28 $\times \;10^{-3}$	−3.57 $\times \;10^{-4}$	−1.41 $\times \;10^{-4}$	**−3.31 × 10^−5^**	1.28 $\times \;10^{-4}$	1.06 $\times \;10^{-6}$	**3.04 × 10^−8^**	1.54 $\times \;10^{-7}$	3.78 $\times \;10^{-8}$	1.32 $\times \;10^{-2}$	9.34 $\times \;10^{-3}$	3.97 $\times \;10^{-4}$	4.17 $\times \;10^{-4}$	**1.97 × 10^−4^**
0.101	5000	1.71 $\times \;10^{-2}$	**3.38 × 10^−4^**	−3.72 $\times \;10^{-3}$	−2.54 $\times \;10^{-3}$	−3.81 $\times \;10^{-3}$	1.81 $\times \;10^{-4}$	6.26 $\times \;10^{-5}$	1.01 $\times \;10^{-5}$	1.07 $\times \;10^{-5}$	**7.70 × 10^−6^**	2.17 $\times \;10^{-2}$	7.92 $\times \;10^{-3}$	4.90 $\times \;10^{-3}$	**4.14 × 10^−3^**	4.71 $\times \;10^{-3}$
	10,000	8.45 $\times \;10^{-3}$	4.62 $\times \;10^{-4}$	−8.81 $\times \;10^{-4}$	**−2.84 × 10^−4^**	−9.03 $\times \;10^{-4}$	1.66 $\times \;10^{-4}$	1.45 $\times \;10^{-5}$	3.49 $\times \;10^{-6}$	**1.66 × 10^−6^**	2.46 $\times \;10^{-6}$	1.54 $\times \;10^{-2}$	3.83 $\times \;10^{-3}$	2.07 $\times \;10^{-3}$	**1.32 × 10^−3^**	1.81 $\times \;10^{-3}$
	20,000	5.92 $\times \;10^{-3}$	1.56 $\times \;10^{-3}$	**7.21 × 10^−4^**	1.04 $\times \;10^{-3}$	1.08 $\times \;10^{-3}$	1.44 $\times \;10^{-4}$	4.20 $\times \;10^{-6}$	8.80 $\times \;10^{-7}$	1.22 $\times \;10^{-6}$	**8.17 × 10^−7^**	1.34 $\times \;10^{-2}$	2.57 $\times \;10^{-3}$	**1.18 × 10^−3^**	1.52 $\times \;10^{-3}$	1.41 $\times \;10^{-3}$
	50,000	4.03 $\times \;10^{-3}$	2.04 $\times \;10^{-3}$	4.91 $\times \;10^{-4}$	**−1.02 × 10^−4^**	5.85 $\times \;10^{-4}$	6.00 $\times \;10^{-5}$	4.31 $\times \;10^{-7}$	1.61 $\times \;10^{-7}$	**1.08 × 10^−7^**	1.26 $\times \;10^{-7}$	8.73 $\times \;10^{-3}$	2.15 $\times \;10^{-3}$	6.34 $\times \;10^{-4}$	**3.44 × 10^−4^**	6.85 $\times \;10^{-4}$
	${1\times 10}^{5}$	3.73 $\times \;10^{-3}$	4.34 $\times \;10^{-3}$	−5.10 $\times \;10^{-4}$	**4.31 × 10^−4^**	−5.17 $\times \;10^{-4}$	9.13 $\times \;10^{-5}$	3.03 $\times \;10^{-7}$	2.83 $\times \;10^{-8}$	4.34 $\times \;10^{-8}$	**2.05 × 10^−8^**	1.03 $\times \;10^{-2}$	4.38 $\times \;10^{-3}$	5.37 $\times \;10^{-4}$	**4.79 × 10^−4^**	5.36 $\times \;10^{-4}$
	${2\times 10}^{5}$	4.38 $\times \;10^{-3}$	5.48 $\times \;10^{-3}$	−5.31 $\times \;10^{-4}$	**−1.61 × 10^−4^**	−3.81 $\times \;10^{-4}$	1.00 $\times \;10^{-4}$	3.41 $\times \;10^{-7}$	**2.22 × 10^−8^**	7.06 $\times \;10^{-8}$	2.88 $\times \;10^{-8}$	1.09 $\times \;10^{-2}$	5.51 $\times \;10^{-3}$	5.52 $\times \;10^{-4}$	**3.11 × 10^−4^**	4.17 $\times \;10^{-4}$

The table shows the bias, variance and root mean squared error of SE estimated from our methods, as compared to the true SE. Sum_Vg, true total heritability explained; *N*, sample size. The best performing method (for estimation of SE) in each scenario is in bold. For other abbreviations, please refer to Table 1.

**Table 3 ijms-25-01347-t003:** Coverage probabilities of different union CI (UCI) approaches for 95% CI.

N	Union CI Type	Coverage (Vg = 0.295)	Coverage (Vg = 0.191)	Coverage (Vg = 0.101)
5000	Standard	0.75	0.97	0.77
	Percentile	1	1	0.78
	Standard + Percentile	1	1	0.78
10,000	Standard	0.6	0.67	0.94
	Percentile	0.99	1	1
	Standard + Percentile	0.99	1	1
20,000	Standard	0.89	0.84	0.96
	Percentile	0.91	1	1
	Standard + Percentile	0.97	1	1
50,000	Standard	1	1	0.9
	Percentile	1	1	1
	Standard + Percentile	1	1	1
$1\times 10^{5}$	Standard	1	1	1
	Percentile	1	1	1
	Standard + Percentile	1	1	1
$2\times 10^{5}$	Standard	0.96	1	1
	Percentile	0.13	0.66	1
	Standard+Percentile	0.96	1	1

Notes for Table 3: The following methods were explored: 1. Normal approximation (standard approach) without bias correction (one estimator) or with bootstrap bias correction (3 estimators), and the union of CIs was taken; (“Standard”). 2. Percentile approach without bias correction (3 estimators) and with bias correction (3 estimators), and the union of CIs was taken; (“Percentile”). 3. Union of the final union CIs obtained from 1 and 2. (“Standard+Percentile”). Coverage refers to the coverage probabilities based on simulations.

**Table 4 ijms-25-01347-t004:** Summary of the immune traits being studied.

Trait	Abbreviation	GWAS ID	*N*	SNP_h2 (LDSC)	SNP_h2_se (LDSC)
Stem cell factor	SCF	ebi-a-GCST004429	8290	−0.06	0.055
Interleukin-4	IL4	ebi-a-GCST004453	8124	−0.0446	0.0595
Interleukin-17	IL17	ebi-a-GCST004442	7760	−0.0407	0.0623
Hepatocyte growth factor	HGF	ebi-a-GCST004449	8292	−0.0311	0.0579
Basic fibroblast growth factor	FGFBasic	ebi-a-GCST004459	7565	−0.0159	0.0597
Stromal cell-derived factor-1 alpha (CXCL12)	SDF1a	ebi-a-GCST004427	5998	−0.0116	0.0713
Interleukin-6	IL6	ebi-a-GCST004446	8189	−0.0071	0.0568
Platelet derived growth factor BB	PDGFbb	ebi-a-GCST004432	8293	−0.0043	0.0624
TNF-related apoptosis inducing ligand	TRAIL	ebi-a-GCST004424	8186	0.0125	0.0613
Interferon-gamma	IFNg	ebi-a-GCST004456	7701	0.0134	0.0624
Granulocyte colony-stimulating factor	GCSF	ebi-a-GCST004458	7904	0.0173	0.0601
Interleukin-10	IL10	ebi-a-GCST004444	7681	0.0186	0.0691

Trait, trait name of analyzed GWAS dataset; abbreviation, abbreviation of the trait name; GWAS ID, ID of GWAS dataset for downloading from the IEU OpenGWAS Project; N, sample size; SNP_h2 (LDSC), SNP heritability estimated by LDSC as reported in GWASAtlas; SNP_h2_se (LDSC), standard error of SNP heritability estimated by LDSC as reported in GWASAtlas.

**Table 5 ijms-25-01347-t005:** SE of the sum of variance explained estimated by different resampling approaches, for 12 immune traits (under different *r^2^* pruning thresholds).

Trait	*N*	LDSC		SumVg							
		h2	se	h2	*r^2^*	n_pruned_snp	se_jack1	se_jack_del_d	se_paraboot	se_fdrboot1	se_fdrboot2
SCF	8290	−0.06	0.055	0.333	0.1	428,593	0.0926	0.0822	0.0679	0.0443	0.0514
				0.185	0.05	251,008	0.0526	0.0456	0.0467	0.0502	0.0517
				0.105	0.025	127,908	0.0307	0.0313	0.0272	0.0397	0.0335
				**0.100**	**0.01**	**61,938**	**0.0310**	**0.0200**	**0.0220**	**0.0252**	**0.0265**
				0.092	0.005	51,370	0.0229	0.0169	0.0201	0.0235	0.0230
				0.101	0.002	48,088	0.0319	0.0153	0.0220	0.0226	0.0198
				0.102	0.001	47,108	0.0316	0.0155	0.0223	0.0216	0.0188
IL4	8124	−0.0446	0.0595	0.503	0.1	427,005	0.1218	0.1133	0.0616	0.0563	0.0569
				0.377	0.05	249,710	0.1000	0.0823	0.0484	0.0445	0.0453
				0.302	0.025	127,248	0.0650	0.0594	0.0318	0.0365	0.0336
				**0.235**	**0.01**	**61,685**	**0.0529**	**0.0313**	**0.0247**	**0.0240**	**0.0236**
				0.215	0.005	51,196	0.0472	0.0278	0.0227	0.0217	0.0221
				0.197	0.002	47,878	0.0571	0.0273	0.0228	0.0225	0.0253
				0.187	0.001	46,911	0.0482	0.0244	0.0198	0.0226	0.0242
IL17	7760	−0.0407	0.0623	0.352	0.1	427,226	0.1240	0.0946	0.0692	0.0625	0.0609
				0.228	0.05	250,259	0.0683	0.0668	0.0499	0.0495	0.0495
				0.299	0.025	127,479	0.0877	0.0568	0.0360	0.0380	0.0323
				**0.234**	**0.01**	**61,756**	**0.0485**	**0.0340**	**0.0239**	**0.0267**	**0.0256**
				0.196	0.005	51,215	0.0475	0.0295	0.0237	0.0190	0.0249
				0.195	0.002	47,887	0.0634	0.0231	0.0231	0.0226	0.0210
				0.188	0.001	46,931	0.0568	0.0242	0.0183	0.0211	0.0215
HGF	8292	−0.0311	0.0579	0.366	0.1	428,318	0.0917	0.0864	0.0569	0.0642	0.0593
				0.242	0.05	250,843	0.0812	0.0722	0.0483	0.0492	0.0491
				0.205	0.025	127,850	0.0657	0.0488	0.0327	0.0326	0.0357
				**0.098**	**0.01**	**61,906**	**0.0379**	**0.0224**	**0.0225**	**0.0260**	**0.0242**
				0.115	0.005	51,301	0.0347	0.0199	0.0224	0.0230	0.0203
				0.111	0.002	47,878	0.0414	0.0162	0.0189	0.0211	0.0215
				0.108	0.001	46,934	0.0312	0.0171	0.0221	0.0208	0.0211
FGFBasic	7565	−0.0159	0.0597	0.269	0.1	427,284	0.0835	0.0902	0.0656	0.0530	0.0577
				0.217	0.05	249,930	0.0891	0.0604	0.0473	0.0504	0.0468
				0.117	0.025	127,587	0.0452	0.0431	0.0340	0.0363	0.0358
				**0.133**	**0.01**	**61,911**	**0.0408**	**0.0301**	**0.0232**	**0.0239**	**0.0275**
				0.135	0.005	51,259	0.0376	0.0243	0.0242	0.0267	0.0219
				0.143	0.002	47,874	0.0362	0.0218	0.0185	0.0233	0.0245
				0.126	0.001	46,914	0.0392	0.0206	0.0227	0.0214	0.0208
SDF1a	5998	−0.0116	0.0713	0.395	0.1	425,165	0.1120	0.1068	0.0731	0.0757	0.0870
				0.256	0.05	248,727	0.0872	0.0750	0.0580	0.0565	0.0631
				0.213	0.025	126,986	0.0707	0.0462	0.0431	0.0472	0.0468
				**0.163**	**0.01**	**61,680**	**0.0497**	**0.0380**	**0.0359**	**0.0349**	**0.0324**
				0.190	0.005	51,092	0.0708	0.0318	0.0250	0.0297	0.0270
				0.165	0.002	47,702	0.0447	0.0270	0.0294	0.0304	0.0301
				0.159	0.001	46,789	0.0512	0.0232	0.0279	0.0258	0.0308
IL6	8189	−0.0071	0.0568	0.422	0.1	427,566	0.0878	0.0896	0.0510	0.0575	0.0594
				0.227	0.05	250,247	0.0620	0.0713	0.0402	0.0463	0.0468
				0.158	0.025	127,503	0.0672	0.0457	0.0372	0.0300	0.0360
				**0.139**	**0.01**	**61,931**	**0.0606**	**0.0258**	**0.0220**	**0.0247**	**0.0220**
				0.114	0.005	51,332	0.0288	0.0176	0.0196	0.0227	0.0236
				0.115	0.002	47,930	0.0302	0.0164	0.0191	0.0226	0.0202
				0.117	0.001	46,944	0.0319	0.0175	0.0227	0.0209	0.0211
PDGFbb	8293	−0.0043	0.0624	0.432	0.1	427,743	0.0907	0.0993	0.0726	0.0653	0.0676
				0.341	0.05	250,325	0.0670	0.0808	0.0600	0.0496	0.0576
				0.307	0.025	127,567	0.0735	0.0554	0.0370	0.0334	0.0326
				**0.154**	**0.01**	**61,789**	**0.0372**	**0.0250**	**0.0213**	**0.0245**	**0.0243**
				0.125	0.005	51,140	0.0310	0.0226	0.0234	0.0230	0.0221
				0.120	0.002	47,822	0.0258	0.0205	0.0214	0.0208	0.0233
				0.117	0.001	46,853	0.0392	0.0192	0.0201	0.0209	0.0226
TRAIL	8186	0.0125	0.0613	0.559	0.1	423,391	0.0613	0.1018	0.0785	0.0790	0.0750
				0.304	0.05	247,717	0.0543	0.1190	0.0526	0.0503	0.0439
				0.242	0.025	126,350	0.0607	0.0647	0.0321	0.0362	0.0370
				**0.128**	**0.01**	**61,114**	**0.0316**	**0.0251**	**0.0242**	**0.0229**	**0.0277**
				0.127	0.005	50,633	0.0298	0.0231	0.0255	0.0239	0.0268
				0.128	0.002	47,359	0.0332	0.0216	0.0233	0.0215	0.0266
				0.121	0.001	46,415	0.0358	0.0195	0.0222	0.0229	0.0256
IFNg	7701	0.0134	0.0624	0.393	0.1	426,740	0.0946	0.0811	0.0528	0.0590	0.0594
				0.241	0.05	249,818	0.0655	0.0628	0.0553	0.0520	0.0509
				0.244	0.025	127,514	0.0734	0.0582	0.0330	0.0406	0.0320
				**0.138**	**0.01**	**61,890**	**0.0289**	**0.0303**	**0.0267**	**0.0239**	**0.0257**
				0.138	0.005	51,314	0.0424	0.0201	0.0222	0.0248	0.0293
				0.141	0.002	47,918	0.0321	0.0204	0.0251	0.0248	0.0286
				0.137	0.001	46,934	0.0253	0.0183	0.0223	0.0246	0.0233
GCSF	7904	0.0173	0.0601	0.246	0.1	427,393	0.0707	0.0820	0.0620	0.0604	0.0580
				0.198	0.05	250,222	0.0636	0.0607	0.0402	0.0436	0.0486
				0.164	0.025	127,583	0.0501	0.0415	0.0302	0.0360	0.0327
				**0.142**	**0.01**	**61,846**	**0.0434**	**0.0257**	**0.0280**	**0.0239**	**0.0257**
				0.122	0.005	51,266	0.0379	0.0196	0.0205	0.0247	0.0238
				0.120	0.002	47,919	0.0413	0.0183	0.0219	0.0236	0.0201
				0.112	0.001	46,939	0.0312	0.0159	0.0234	0.0207	0.0202
IL10	7681	0.0186	0.0691	0.331	0.1	427,218	0.0621	0.1019	0.0584	NA	NA
				0.310	0.05	250,109	0.0670	0.0858	0.0448	NA	NA
				0.198	0.025	127,543	0.0566	0.0463	0.0356	0.0382	0.0406
				**0.130**	**0.01**	**61,944**	**0.0328**	**0.0225**	**0.0251**	**0.0268**	**0.0258**
				0.141	0.005	51,257	0.0400	0.0220	0.0237	0.0282	0.0238
				0.148	0.002	47,880	0.0433	0.0183	0.0204	0.0271	0.0231
				0.142	0.001	46,898	0.0317	0.0194	0.0219	0.0261	0.0228

This table shows the estimated total SNP-based heritability and their SEs for 12 immune traits. We also show a comparison of the estimates between LDSC and SumVg. Trait, N, LDSC (h2, se) have the same meaning as in Table 4; h2, heritability estimated by SumVg across a set of *r*^2^ pruning thresholds; *r*^2^, the *r*^2^ pruning threshold; n_pruned_snp, number of SNPs after LD pruning at the corresponding *r^2^* threshold; se_jack_1, se_jack_del_d, se_paraboot, se_fdrboot1 and se_fdrboot2 are SE estimated by different approaches as described above; “NA” was shown when “locfdr” failed to estimate local false discovery rate. The estimates with *r*^2^ = 0.01 were highlighted, as we observed that in general the heritability estimates stabilize at *r*^2^ ~ 0.01.

## Data Availability

The GWAS summary statistics of the immune traits are downloaded from the GWASAtlas (https://atlas.ctglab.nl/; accessed 1 May 2023); LDSC heritability were extracted from the same website.

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
