# Peer review of "SumVg: Total Heritability Explained by All Variants in Genome-Wide Association Studies Based on Summary Statistics with Standard Error Estimates"

_ijms, 2024, doi:10.3390/ijms25021347_

Round 1
Reviewer 1 Report
Comments and Suggestions for Authors
My comments are included in the attached file.

Reviewer 2 Report
Comments and Suggestions for Authors
This is an interesting study that report a self-developed program to estimate the SE of SNP-based heritability from GWAS summary statistics by comparing several resampling-based approaches. They have also validated their method by applying the program to estimate SNP-based 22 heritability of 12 immune-related traits (levels of cytokines and growth factors). The results seems consistent and convincing, which could be useful for researchers in the field. I only suggest that the main results from Table 2&4 could be better represented and illustrated in more visually effective figures?
Reviewer 3 Report
Comments and Suggestions for Authors
My comments are in the attached word file.

Comments on the Quality of English LanguageThe paper has minor typographical errors which needs to be corrected.
Round 2
Reviewer 3 Report
Comments and Suggestions for Authors
I thank the authors for addressing my concerns. I only have one comment. In the revision letter the authors have agreed to several points that I highlighted as limitations to their current approach. However, I did not see them mentioned in the discussion section of the revised manuscript. Can the authors clearly mention all the current limitations (in a paragraph or two) in their discussion section so that the readers/users become aware of them.
Other than the above corrections, I have no further comment.
